# Curine Inhibits Macrophage Activation and Neutrophil Recruitment in a Mouse Model of Lipopolysaccharide-Induced Inflammation

**DOI:** 10.3390/toxins11120705

**Published:** 2019-12-03

**Authors:** Jaime Ribeiro-Filho, Fagner Carvalho Leite, Andrea Surrage Calheiros, Alan de Brito Carneiro, Juliana Alves Azeredo, Edson Fernandes de Assis, Celidarque da Silva Dias, Márcia Regina Piuvezam, Patrícia T. Bozza

**Affiliations:** 1Laboratório de Investigação em Genética e Hematologia Translacional, Instituto Gonçalo Moniz, FIOCRUZ, Salvador 40296-710, Brazil; 2Laboratório de Imunofarmacologia, Departamento de Fisiologia e Patologia, UFPB, João Pessoa 58051-900, Brazil; fagnercarvalho.farm@gmail.com (F.C.L.); mrpiuvezam@ltf.ufpb.br (M.R.P.); 3Laboratório de Imunofarmacologia, Instituto Oswaldo Cruz, FIOCRUZ, Rio de Janeiro 21040-360, Brazil; andrea.surrage@gmail.com (A.S.C.); alan.fiocruz@gmail.com (A.d.B.C.); jazeredo@ioc.fiocruz.br (J.A.A.); edassis@ioc.fiocruz.br (E.F.d.A.); pbozza@gmail.com (P.T.B.); 4Laboratório de Fitoquímica, Departamento de Ciências Farmacêuticas, UFPB, João Pessoa 58051-900, Brazil; celidarquedias@ltf.ufpb.br

**Keywords:** Curine, alkaloid, macrophage, neutrophil, lipopolysaccharide

## Abstract

Curine is a bisbenzylisoquinoline alkaloid (BBA) with anti-allergic, analgesic, and anti-inflammatory properties. Previous studies have demonstrated that this alkaloid is orally active at non-toxic doses. However, the mechanisms underlying its anti-inflammatory effects remain to be elucidated. This work aimed to investigate the effects of curine on macrophage activation and neutrophil recruitment. Using a murine model of lipopolysaccharide (LPS)-induced pleurisy, we demonstrated that curine significantly inhibited the recruitment of neutrophils in association with the inhibition of cytokines tumor necrosis factor (TNF-α), interleukin (IL)-1β, IL-6, monocyte chemotactic protein (CCL2/MCP-1) as well as leukotriene B_4_ in the pleural lavage of mice. Curine treatment reduced cytokine levels and the expression of iNOS in in vitro cultures of macrophages stimulated with LPS. Treatment with a calcium channel blocker resulted in comparable inhibition of TNF-α and IL-1β production, as well as iNOS expression by macrophages, suggesting that the anti-inflammatory effects of curine may be related to the inhibition of calcium-dependent mechanisms involved in macrophage activation. In conclusion, curine presented anti-inflammatory effects that are associated with inhibition of macrophage activation and neutrophil recruitment by inhibiting the production of inflammatory cytokines, LTB_4_ and nitric oxide (NO), and possibly by negatively modulating Ca^2+^ influx.

## 1. Introduction

Macrophages work as detectors of inflammatory signals, including those produced by the host and the derived from microorganisms, such as lipopolysaccharide (LPS) [1]. LPS signaling through TLR_4_ induces macrophage activation by regulating intracellular pathways involved in cytokine, lipid mediator and oxygen reactive species (ROS) production, in a process regulated by calcium signaling [2,3]. The mediators released by activated macrophages play critical roles in neutrophil recruitment and activation, and therefore, influence the progress of immune responses as well as the development of many inflammatory diseases [4,5].

Curine (Figure 1A) is the principal bisbenzilisoquinoline alkaloid (BBA) obtained from *Chondrodendron platyphyllum* (Menispermaceae). Earlier studies reported that this alkaloid, as well as the structurally related compounds isocurine and 12-O-metilcurine, have promising pharmacological effects [6] corroborating ethnopharmacological data which points *C. phatyphyllum* as a plant with medicinal properties [7]. Studies have shown that BBA are bioactive natural compounds presenting anti-inflammatory, anti-allergic, and analgesic activities [8] and there is evidence that their mechanism of action involves a direct inhibition of calcium channels [6,9,10].

Studies carried out by our group have demonstrated the effects of curine treatment in an experimental model of allergic asthma. The oral administration of this compound to allergic mice significantly inhibited eosinophilic inflammation and airway hyper-responsiveness (AHR), which are critical hallmarks of the allergic response in this model. In addition, curine prevented lipid body formation and cytokine production in vivo, suggesting that it has an inhibitory role in eosinophil activation. A similarity between the anti-allergic effects of curine and verapamil (a calcium channel blocker) as well as inhibition of calcium-induced tracheal contraction by curine strongly suggested that its anti-allergic effects are associated with modulation of calcium-dependent responses [11]. These findings were affirmed by another study in which we demonstrated the anti-allergic effects of curine and verapamil in a mice model of mast cell activation. In addition to inhibiting the scratching behavior, the oral treatment with curine prevented the anaphylactic shock reaction in systemically-challenged mice. Additionally, these treatments inhibited the production of lipid mediators and cytokines associated with mast cell activation [12]. Importantly, an analysis of physical, behavioral, histological, hematologic and biochemical parameters revealed that the oral treatment with curine for seven consecutive days did not induce evident toxicity in mice [11]. Additionally, this alkaloid presented analgesic effects that were not associated with an activity in the central nervous system but involve anti-inflammatory mechanisms [13].

Accumulating evidence places curine as a potent anti-inflammatory and anti-allergic compound with low-toxicity. Through both in vivo and in vitro studies, we have described the general pharmacological properties of this alkaloid. However, the mechanisms underlying its anti-inflammatory effects are still poorly understood. Therefore, the objective of this study was to investigate the effects of curine on macrophage activation and neutrophil recruitment in a mouse model of LPS-induced inflammation. Here we analyze the impact of inflammatory mediator production modulation as well as the importance of calcium influx inhibition in the anti-inflammatory mechanisms of curine.

## 2. Results

### 2.1. Curine Inhibits Neutrophil Recruitment in LPS-Challenged Mice

An intrapleural administration of LPS was observed to induce a significant increase in the number of neutrophils in the pleural lavages of C57Bl/6 mice (Figure 1B). The oral treatment with curine (2.5 mg/kg) or dexamethasone (2 mg/kg) 1 h prior to the LPS challenge caused a significant reduction in neutrophil counts (Figure 1B) in comparison with the group of untreated and challenged mice, thus demonstrating the inhibitory role played by curine with regard to neutrophil recruitment during the pleural inflammation.

### 2.2. Curine Inhibits Inflammatory Mediator Production in Vivo

Based on our finding of increased neutrophil recruitment in response to the LPS challenge, we analyzed the effect of curine treatment on the production of mediators involved in neutrophil recruitment and inflammation. Supernatants obtained from the pleural lavages of LPS-challenged mice presented increased levels of inflammatory mediators (Figure 2A–F), in comparison to unstimulated animals. Curine treatment was observed to significantly inhibit the production of interleukin (IL)-6, tumor necrosis factor (TNF)-α, monocyte chemotactic protein (MCP)-1/CCL2, keratinocyte-derived chemokine (KC/CXCL1) and leukotriene B_4_ (LTB_4_), thereby providing evidence of a link between the inhibitory effect of curine on neutrophil recruitment and associated inflammatory mediator production.

### 2.3. Curine Inhibits Macrophage Activation in Vitro

As activated macrophages are crucially involved in the production of mediators in early inflammatory events, we analyzed the direct effects of curine on macrophage activation by investigating its interference on cytokine production in vitro. Figure 3 shows that stimulation of peritoneal macrophage cultures with LPS increased the levels of IL-1β, IL-6 and TNF-α in comparison with control cells. Pre-treatment with curine at 1 or 10 µM significantly reduced the levels of these cytokines in the supernatants indicating that this BBA inhibits TLR-4 mediated macrophage activation in vitro (Figure 3).

### 2.4. Effects of Calcium Influx Inhibition on Macrophage Activation

Our group recently demonstrated that curine and verapamil presented anti-allergic effects that might be associated with calcium signaling modulation [11,12]. To evaluate the importance of calcium influx inhibition on macrophage activation, as well as its potential participation in curine anti-inflammatory mechanisms, we made a comparison between the effects of curine and verapamil on macrophage activation. As shown in Figure 4, treatment with curine or verapamil at the same concentration induced a similar inhibition in IL-1β (A) and TNF-α (B) production, which suggests that the effects of curine on macrophage activation might be dependent on calcium influx inhibition.

### 2.5. Curine Inhibits Nitric Oxide (NO) Production by Regulating iNOS Expression in Macrophages

It has been demonstrated that LPS stimulates the synthesis of NO via IL-1β, TNF-α, and IFN-γ [14]. Figure 5A illustrates the effects of curine on NO production. The supernatants of LPS-stimulated peritoneal macrophages presented significantly increased concentrations of nitrite, which was significantly decreased by curine treatment. Moreover, while LPS stimulation was found to induce increased expression of iNOS by macrophages, treatment with curine or verapamil reduced the expression of this enzyme (Figure 5B). These findings suggest that the inhibition of NO production through the curine-mediated regulation of iNOS expression could be associated with calcium influx inhibition.

## 3. Discussion

The bisbenzylisoquinoline alkaloids (BBA) constitute a group of secondary metabolites that exert numerous biological effects. The medicinal properties of BBA-rich plants and isolated compounds have been demonstrated in different experimental models, indicating that this class of substances presents promising anti-allergic and anti-inflammatory activities [15,16]. Our group found that curine, a BBA identified as the main constituent of *Chondrodendron platyphyllum* (Menispermaceae), is an orally active alkaloid with potent immunomodulatory effects and low toxicity, which therefore makes it a promising candidate in the development of new anti-inflammatory drugs [16].

In a worldwide context, questions have been raised concerning the efficacy and safety of currently available medications [17]. Although corticosteroids, non-steroidal anti-inflammatory drugs (NSAIDs) and other conventional drugs effectively relieve most inflammatory symptoms, in specific conditions these are not effective, or can cause significant side effects [18]. Accordingly, the development of novel, safe and effective drugs is imperative to improving anti-inflammatory therapy.

Although inflammatory diseases differ in various aspects, some evidence has consistently shown that macrophages and neutrophils perform essential functions in the initiation and development of many inflammatory conditions [19]. The present mouse model of LPS-induced inflammation, used to characterize the effects of curine on macrophage activation and neutrophil recruitment, demonstrated new anti-inflammatory properties of this alkaloid compound. Our findings indicate that orally administered curine inhibited the recruitment of neutrophils to the pleural cavity of LPS-challenged mice. Accordingly, curine treatment reduced levels of IL-6, TNF-α, CCL2/MCP-1, and LTB_4_ in the pleural lavages of these animals, providing evidence of a link between the inhibitory effect of this alkaloid on neutrophil in association with the production of inflammatory mediators.

Neutrophils can rapidly migrate to sites of inflammation [20] in response to inflammatory signals such as chemokines and cytokines produced by resident cells [1]. The chemokine CXCL1 (also known as KC in mice) plays a critical role in neutrophil recruitment and activation by signaling via CXCR2 on these cells [21]. Previous studies have also demonstrated that LTB_4_ acts as an essential chemotactic agent [22,23] by stimulating the recruitment of neutrophils via BLT_1_ receptor activation [24]. It follows that the inhibitory effect of curine on KC/CCL1 and LTB_4_ production might therefore directly impact neutrophil recruitment. Additionally, LPS-induced cytokines, including TNF-α and IL-β, can directly affect neutrophil recruitment by stimulating the expression of adhesion molecules, including selectins and integrins [25,26,27].

As activated macrophages are one of the most critical sources of mediator production in the early phase of inflammation [28], we hypothesized that the inhibition of neutrophil recruitment and cytokine production in the pleural lavage induced by curine might be associated with decreased macrophage activation. Our data show that the production of IL-6, IL-β, and TNF-α was inhibited in murine macrophages stimulated with curine in vitro, which indicates that this compound may regulate neutrophil recruitment by inhibiting the production of key inflammatory mediators in macrophages. On the other hand, the release of products involved in monocyte/macrophage influx and activation, such as MCP-1, by neutrophils can also affect macrophage function [29]. Here, MCP-1 production was found to be significantly inhibited by curine, which suggests that cross-talk between neutrophils and macrophages could be impaired by curine treatment.

Our previous work has demonstrated that curine can exert anti-allergic effects associated with the inhibition of calcium influx [11,12,16]. Using tracheal rings preparations kept in Ca^2+^-free medium, depolarized with KCl and stimulated with cumulative addition of Ca^2+^, Ribeiro-Filho and colleagues [11] demonstrated ex-vivo that curine pre-treatment significantly inhibited calcium-induced trachea contractile response, suggesting that curine inhibits the influx of calcium via blockade of voltage-dependent Ca^2+^ channels in the rat tracheal smooth muscle. In addition, Medeiros and collaborators [10] demonstrated that curine decreased intracellular Ca^2+^ transients in A7r5 cells, which indicated that this alkaloid can have a direct inhibitory effect on L-type Ca^2+^ channels in vascular smooth muscle cells. Here we hypothesized that calcium influx inhibition would impair macrophage activation, which could be partially responsible for the anti-inflammatory effects associated with curine treatment. To confirm this, we compared the impact of verapamil and curine on cytokine production by LPS-stimulated macrophages, as a parameter to evaluate macrophage activation. When administered under identical conditions (concentrations and time of pre-treatment), verapamil and curine demonstrated similar inhibitory effects, suggesting that the modulation of calcium influx is indeed a potential mechanism by which curine inhibits the inflammatory response.

Curine was also found to significantly decrease nitrite concentrations in the supernatants of macrophages stimulated with LPS, indicating that nitric oxide (NO) production was inhibited in vitro. This finding adds to the role played by curine in macrophage activation, since NO production is a hallmark of activated macrophages [14]. Importantly, recent reports have demonstrated that NO, in association with other ROS, is critically involved in neutrophil extracellular trap (NET) formation [30]. Curiously, it was also recently reported that neutrophils might participate in the resolution of inflammation mediated by reparative macrophages [5]. It has been well established that LPS and inflammatory cytokines stimulate NO production by inducing iNOS expression [31]. Here, we demonstrated that curine and verapamil inhibited the expression of iNOS in LPS-stimulated murine macrophages, which thereby provides evidence of the inhibitory effect of these drugs on NO production. This finding further corroborates the impact of curine on TNF-α and IL-1β production and lends support to the notion that curine negatively modulates macrophage activation through the inhibition of a calcium-dependent response. In fact, studies have shown that calcium-dependent signaling potentiates macrophage activation [32] and stimulates proinflammatory cytokine production [33] in these cells. Therefore, further studies addressing the role of curine in modulating calcium influx and associated signaling pathways in immune cells, including macrophages, will contribute to characterize the molecular mechanism of action of curine as an anti-inflammatory compound.

We previously reported that a single dose of curine administered orally as pre-treatment exhibited anti-inflammatory and analgesic effects in mice [13]. Our investigation of the analgesic effects of curine revealed that instead of acting by way of neurogenic mechanisms, curine acts through anti-inflammatory mechanisms associated with the inhibition of PGE_2_ production. These findings are in line with our previous work which demonstrated that curine inhibited the synthesis of cysteinyl leukotrienes and PGD_2_ by mast cells [12]. Taken together, these findings indicate that curine affects signaling pathways involved in lipid mediator synthesis, possibly due to interference in leukocyte activation. Finally, the present findings suggest that the anti-inflammatory and analgesic effects of curine are also related to the inhibition of critical mediators of inflammatory pain, including IL-1β, TNF-α, and NO [34,35,36].

In conclusion, curine was shown to exert anti-inflammatory effects associated with the inhibition of macrophage activation and neutrophil recruitment, as well as reduced production of inflammatory cytokines, LTB4, NO and the modulation of Ca^2+^ influx (Figure 6).

## 4. Materials and Methods

### 4.1. Preparation of Curine Solution

Curine was purified from the total tertiary alkaloid fraction (TTA) obtained from the root bark of *Chondrodendron platyphyllum* Hil St. (Miers) as previously described [13]. The TTA was submitted to column chromatography followed by thin-layer chromatography (TLC) purification, from which curine was obtained in the form of a crystal. The chemical structure was analyzed by spectroscopy and comparison with the literature data [37]. The purity of curine was analyzed by NMR ^13^C and NMR ^1^H (CDCl_3_, 400 MHz) data from the crystals when compared to the literature data [13] and the substance was considered spectroscopically pure. After purification, 1 mg of the crystal was dissolved in 50 μL of 1 N HCl and 500 μL of distilled water. The pH was adjusted (between 7–8) with 1 N NaOH and volume was adjusted to 1000 μL, with dilutions performed in phosphate-buffered saline (PBS). The use of *C. platyphyllum* in the present study was registered in the National System of Genetic Heritage Management and Associated Traditional Knowledge (SisGen, protocol A84A87E).

### 4.2. Animals

Male C57Bl/6 mice (age 6–8 weeks) weighing 20–30 g obtained from the Oswaldo Cruz (Fiocruz, Rio de Janeiro, Brazil) were maintained with food and water ad libitum in cages at room temperature ranging from 22 to 24 °C under a 12 h light/dark cycle. This study was carried out in accordance with the recommendations established by the Brazilian National Council for the Control of Animal Experimentation (CONCEA). All experimental protocols were approved by the Animal Welfare Committee of the Oswaldo Cruz Foundation (CEUA/FIOCRUZ-RJ, protocol #L-002/08).

### 4.3. Treatments

For in vivo experimentation, animals (6–8 per group) were randomly assigned by body weight for single oral pre-treatment with curine (2.5 mg/kg), dexamethasone (2 mg/kg) or PBS (negative control). All treatments were performed by simple awake gavage [38,39] 1 h prior to LPS challenge. Briefly, treatments (0.1 mL/10 g body weight) were administered using a 31 mm long, 1 mm diameter reusable curved stainless-steel feed needle containing a 1.7 mm ball at the tip (Bronther, Ribeirão Preto, SP, Brazil). The gavage needle was gently inserted into the oral cavity, ensuring the correct passage through the esophagus. For in vitro experiments, cells were treated with curine or verapamil 1 or 10 μM) or PBS 1 h before stimulation. Curine dosage (2.5 mg/kg) was based on results obtained by Ribeiro-Filho and colleagues [11].

### 4.4. LPS-Induced Pleurisy

Male C57Bl/6 mice (*n* = 6–8) were orally pre-treated with curine (2.5 mg/kg) or dexamethasone (2 mg/Kg) 1 h prior to the pleurisy protocol [40]. Animals were anesthetized with isoflurane (Forane™, Abbott, São Paulo, SP, Brazil) and challenged through an intrathoracic (i.t) injection of LPS (250 ng/cavity) dissolved in 100 μL of PBS. A group of mice receiving the same volume of PBS was used as the control. Four hours following the LPS injection, the animals were euthanized by CO_2_ and the pleura was surgically exposed. Pleural lavage was collected by washing the pleural cavity with 1 mL of heparinized PBS (20 U/mL).

### 4.5. Leukocyte Counting

Leukocytes were counted under light microscopy after diluting the pleural lavage samples in Turk fluid (2% acetic acid). Differential counts were performed under an objective lens at 100× magnification after staining by the May–Grunwald–Giemsa method.

### 4.6. Peritoneal Macrophage Cultures

Peritoneal macrophages from C57Bl/6 mice were obtained four days after the injection of 4% thioglycollate. The peritoneal cavity was washed with RPMI 1640 medium supplemented with 100 U/mL penicillin and 100 μg/mL streptomycin (Thermo Fisher Scientific, Waltham, MA, USA). Cells were adjusted to 2 × 10^6^/mL and plated on 24-well culture plates (500 μL) at 37 °C under 4% CO_2_ overnight. Following incubation, cells were pre-treated with curine, or alternatively with verapamil, (both at 1 or 10 μM) and stimulated with LPS (500 ng/mL) 1 h later. Of note, curine pre-treatment (at either 1 or 10 μM) did not affect cell viability, which was higher than 90% in all experiments.

### 4.7. Cytokine and LTB_4_ Analysis

Samples of the pleural lavage were centrifuged at 500 g for 8 min at 4 °C to obtain the supernatants. The concentrations of IL-1β, IL-6, TNF-α, CCL2/MCP-1 and KC/CXCL-1 in these supernatants were determined using a multiplex fluorescent microbead immunoassay (Bio-Rad Laboratories, Hercules, CA, USA). Cytokine levels were quantified using a Luminex technology (Bio-Plex Workstation; Bio-Rad Laboratories, Hercules, CA, USA). Data analysis was performed using Bio-Plex software (Bio-Rad Laboratories, Hercules, CA, USA). The concentrations of LTB_4_ in the pleural lavages, as well as cytokines in the supernatants of macrophage cultures, were analyzed using ELISA kits in accordance with the manufacturer’s instructions (Cayman Chemical, Ann Arbor, MI, USA: LTB_4_ and R&D Systems, Minneapolis, MN, USA: Cytokines). All analyses were performed 4 h after stimulation with LPS.

### 4.8. Nitrite Quantification

For NO_2_^−^ determination, 100 μL of macrophage culture supernatant was removed 24 h after LPS stimulus and incubated with Griess reagent (1% sulfanilamide and 0.1% naphthylenediamine hydrochloride in 2.5% H_3_PO_4_) for 10 min at room temperature. The readings were performed at 540 nm using a spectrophotometer (Titertek Multiscan, Flow Laboratories, Eflab Oy, Helsinki, Finland). Nitrite concentrations were calculated using a standard reference curve obtained from NaNO_2_ (1–200 μM in culture medium).

### 4.9. SDS-PAGE and Western Blotting

Eighteen hours after LPS stimulus, Western blot was used to analyze iNOS expression. Briefly, cells were washed in PBS buffer and homogenized with 10 mM Tris-HCl buffer (pH 7.4), 150 mM NaCl, 0.5% triton X-100, 10% glycerol (*v*/*v*), 0.1 mM EDTA, 1 mM Dithiothreitol (DTT) and a cocktail of protease inhibitors (Roche Diagnostics GmbH, Mannheim, Germany). Proteins from the cell homogenate were separated by polyacrylamide gels in the presence of 10% SDS at a constant current of 16 mA. Full-range rainbow molecular weight markers (RPN800E, GE Healthcare Life Sciences, Piscataway, NJ, USA) were used as a relative molecular mass standard. After running the gels, samples were transferred at 200 mA (2.7 mA/cm^2^) to a nitrocellulose membrane using 25 mM Tris–HCl and 192 mM glycine, pH 8.3, at 4 °C for 120 min. The membranes were then blocked with Tris-buffered saline (TBS)-0.5 tween 20 and 5% milk for 1 h at room temperature, incubated with a polyclonal antibody (1:1000) against iNOS (BD-610333) for 18 h at 4 °C, followed by incubation with a secondary antibody (anti-rabbit IgG-HRP, PI.1000, Vector Laboratories) for 1 h at room temperature. Reactions were developed using a Super Signal West Pico Chemiluminescent Substrate (Thermo Fisher Scientific, Rockford, IL, USA).

### 4.10. Statistical Analyses

Data were analyzed by one-way ANOVA followed by Tukey’s post-test using GraphPad Prism software version 5.02 (GraphPad, San Diego, CA, USA, 2016). Values of in vivo experiments are expressed as means ± SD and values of in vitro assays as means ± Standard Error of Mean (SEM). Statistical significance was considered when *p* < 0.05.

## Figures and Tables

**Figure 1 toxins-11-00705-f001:**
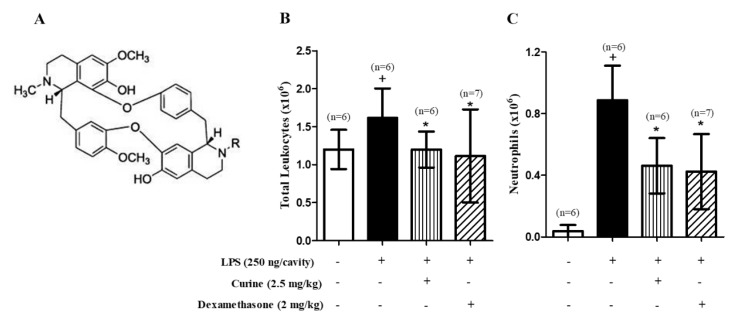
Effect of curine on neutrophil recruitment in lipopolysaccharide (LPS)-induced pleurisy. (**A**) The chemical structure of curine. Total leukocytes (**B**) and neutrophils (**C**) per pleural lavage of C57Bl/6 mice orally pre-treated with curine (2.5 mg/kg) or dexamethasone (2 mg/kg), counted under light microscopy 4h after LPS-challenge. Results are expressed as means ± SD from at least six animals. + significant difference (*p* < 0.05) from the unchallenged group; * significant difference (*p* < 0.05) from the untreated LPS-challenged group. Statistical significance was determined with one-way ANOVA and post hoc Tukey test.

**Figure 2 toxins-11-00705-f002:**
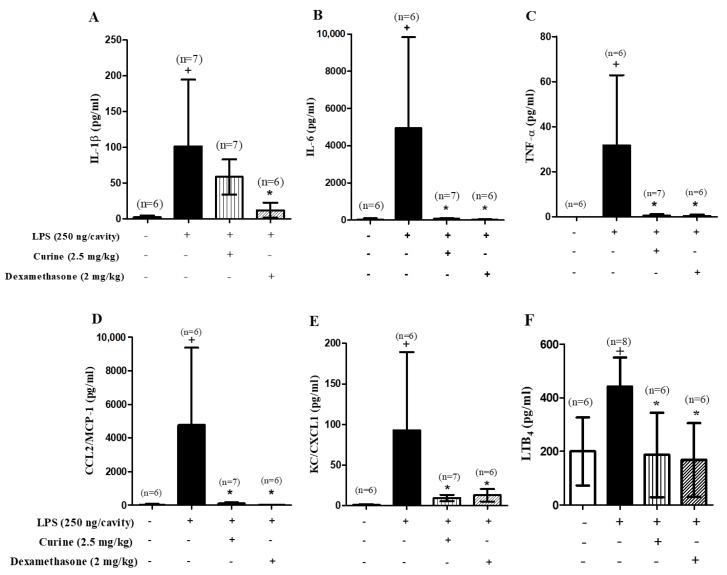
Effects of curine pre-treatment on in vivo cytokine production 4h after LPS challenge. Concentrations of interleukin (IL)-1β **(A**), IL-6 (**B**), tumor necrosis factor (TNF)-α (**C**), CCL2/monocyte chemotactic protein (MCP)-1 (**D**), keratinocyte-derived chemokine (KC/CXCL-1) (**E**) and leukotriene B_4_ (LTB_4_)(**F**) in the pleural lavages of C57Bl/6 mice orally pre-treated with curine (2.5 mg/kg) or dexamethasone (2 mg/kg). These results are expressed as the mean ± SD of at least 6 animals. + significant difference (*p* < 0.05) from the unchallenged group; * significant difference (*p* < 0.05) from the untreated LPS-challenged group. Statistical significance was determined with one-way ANOVA and post hoc Tukey test. Assay range of IL-1β: 10.36–60,631 pg/mL; IL-6: 0.74–12,053 pg/mL; TNF-α: 5.86–59,626 pg/mL; CCL2/MCP-1: 22.4–41,873 pg/mL; KC/CXCL1: 3.2–182 pg/mL and LTB_4_: 3.9–500 pg/mL.

**Figure 3 toxins-11-00705-f003:**
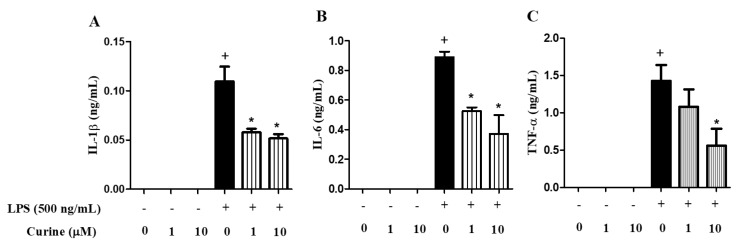
Effects of curine treatment on macrophage activation 4 h after LPS challenge. Concentrations of IL-1β (**A**), IL-6 (**B**) and TNF-α (**C**) in the supernatants of peritoneal macrophage cultures treated with curine (1 or 10 μM) 4h after the stimulus with LPS (500 ng/mL). Results are expressed as means ± Standard Error of Mean (SEM) of two experiments performed in triplicate. + significant difference (*p* < 0.05) from the unchallenged cells; * significant difference (*p* < 0.05) from the untreated LPS-challenged cells. Statistical significance was determined with one-way ANOVA and post hoc Tukey test. Assay range of IL-1β and IL-6: 15.6–1000 pg/mL and TNF-α: 31.3–2000 pg/mL.

**Figure 4 toxins-11-00705-f004:**
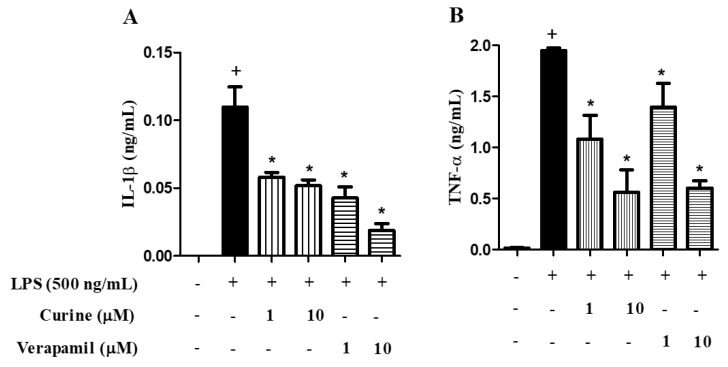
Effects of calcium influx inhibition on macrophage activation. Concentrations of IL-1β (**A**), TNF-α (**B**) in the supernatants of peritoneal macrophage cultures treated with curine or verapamil (1 or 10 μM) were evaluated 4 h after LPS challenge. Results are expressed as means ± SEM from two experiments performed in triplicate. + significant difference (*p* < 0.05) from the unchallenged cells; * significant difference (*p* < 0.05) from the untreated LPS-challenged cells. Statistical significance was determined with one-way ANOVA and post hoc Tukey test. Assay range of IL-1β: 15.6–1000 pg/mL and TNF-α: 31.3–2000 pg/mL.

**Figure 5 toxins-11-00705-f005:**
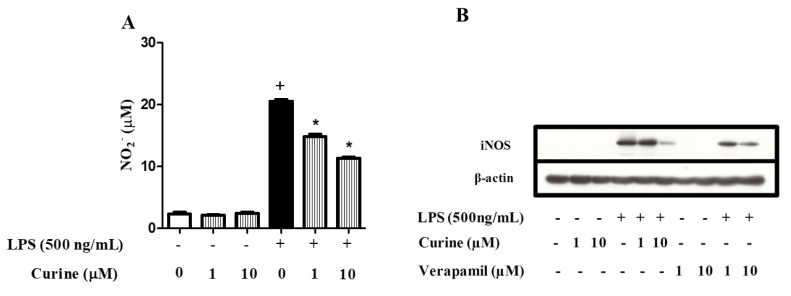
Effects of curine on nitric oxide (NO) production and iNOS expression. Concentrations of nitrite (**A**) in the supernatants of peritoneal macrophages treated with curine or verapamil (1 or 10 μM) 24 h after LPS challenge. iNOS expression (**B**) was analyzed by Western blotting 18 h after LPS stimulus. Results are expressed as means ± SEM from two experiments performed in triplicate. + significant difference (*p* < 0.05) from the unchallenged cells; * significant difference (*p* < 0.05) from the untreated LPS-challenged cells. Statistical significance was determined with one-way ANOVA and post hoc Tukey test.

**Figure 6 toxins-11-00705-f006:**
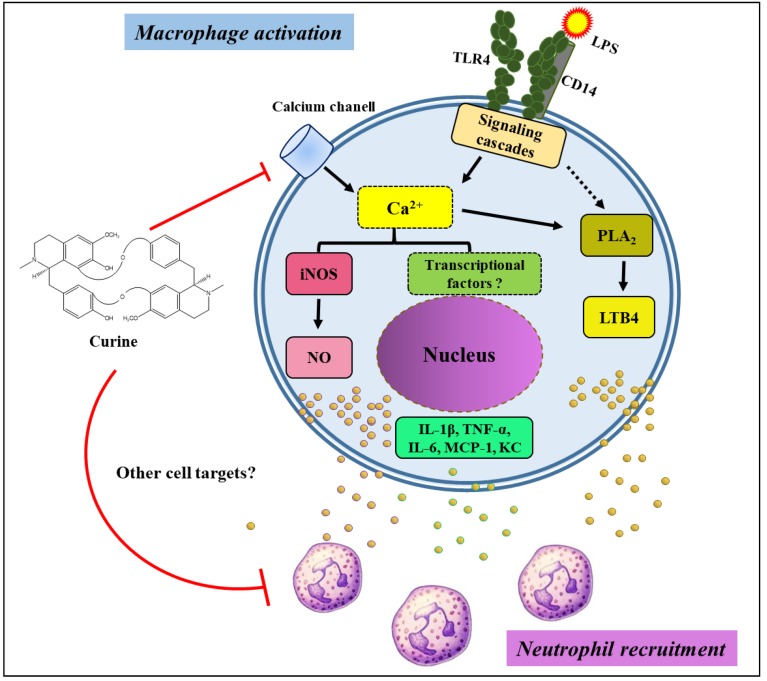
Schematic diagram illustrating a potential mechanism of action by curine on macrophage activation and neutrophil recruitment. The inhibition of calcium influx could be affecting signaling pathways associated with the synthesis of NO, LTB_4_ and cytokines, which could impair neutrophil recruitment. Importantly, the possibility that curine may interfere with other cell targets cannot be ruled out.

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
