# Peer review of "Curine Inhibits Macrophage Activation and Neutrophil Recruitment in a Mouse Model of Lipopolysaccharide-Induced Inflammation"

_toxins, 2019, doi:10.3390/toxins11120705_

Round 1
Reviewer 1 Report
The main aims of this manuscript is to evaluate the anti-inflammatory mechanism of curine in LPS-challenged mice and in vitro macrophages. The results show that curine may exhibit anti-inflammatory effects by inhibiting the activation of macrophage and the recruitment of neutrophils. This research is well-conducted, but there are a few concerns as below.
This manuscript needs language and scientific editing. In the text and figure captions, the concentration of Dexamethasome should be 2 mg/kg, but not 2.5 mg/kg? Please check it throughout the manuscript. Please clearly define the statistic difference (p < ?) in each figures.
Author Response
Response to Reviewer 1 Comments
Point 1: This manuscript needs language and scientific editing. In the text and figure captions, the concentration of Dexamethasone should be 2 mg/kg, but not 2.5 mg/kg? Please check it throughout the manuscript. Please clearly define the statistic difference (p < ?) in each figures.
Response 1: Thank you for your contribution. The manuscript has been completely revised, and the changes are highlighted in the text. Dexamethasone dose was corrected to 2 mg/kg, and statistical difference (p <0.05) was defined in each figure.

Reviewer 2 Report
In their manuscript, the authors set out to investigate the "effects of curine on macrophage activation and neutrophil recruitment" in a murine LPS-induced pleurisy model. Furthermore, they want to analyze the importance of calcium influx inhibition for the effects of curine. Although many of these objectives are addressed adequately, there a some major issues remaining, which preclude publication of the current manuscript version:
MAJOR:
Methods must be described in more detail!A statement on the purity of the curine used, and how purity has been determined, is absolutely essential and missing. Furthermore, neither the age of the mice is mentioned, nor the procedure of allocating the mice into the experimental groups is detailed (randomization?). The procedure of "oral pre-treatment" must be specified: Details and Reference for the technique? Simple oral gavage? Which size? The protocol and anesthetic procedures used for LPS pleurisy should also be detailed or referenced.
The lower limit of quantitation for the individual cytokines must be stated in order to allow for a correct interpretation of the data.
Figures:a) A short description of how significance has been established should be contained within each figure caption (e.g. * significant difference (p<0.05; determined with one-way ANOVA and post hoc Tukey test.).
b) Since group sizes are varying, SEM must be avoided, because it depends on the group size. In order to depict the group size in the figures, data must either be show in the highest possible granularity, e.g. as scatter plots, or the number of individuals per group must be shown near the respective bar of the graph. Did you check for cytokine concentrations in the sera of the mice, too? Can you comment on how sure you are about not having contaminated pleural lavages with blood/ serum components? A major drawback of the study is that it claims to show an impact of curine on calcium / calcium flux, but fails to do so, because no corresponding data is presented: No measurement of intracellular calcium concentrations or calcium fluxes have been done! Only indirect data indicating a mechanism similar to verapamil is presented. Thus, conclusions with regard to the molecular mechanism of curine must be tempered, or data on calcium influx / concentrations must be presented.
MINOR:
How many mice were used for the isolation of macrophages? How have the supernatants of pleural lavages been prepared? Have all procedures of the Western Blotting been performed at 4°C? The 200 mA in blotting correspond to how many mA/cm²? Why did you measure nitrite and not "total NO" corresponding to the sum of nitrite and nitrate? Please correct NO into NO2- in 4.8 and NO-2 into NO2- in figure 5. Please stick to SI units: it's kg, not Kg. What was the rationale behind using 500 ng/mL LPS? Did you try other concentrations of LPS before? The systematic name CCL2 should be mentioned alongside MCP-1. Figure 1: Are these the total counts per BAL? Please correct the y-axis labelling: It should be x106 not x10-6. Please be more specific in l. 105: I fully agree that you showed “that this BBA inhibits [TLR-4 mediated] macrophage activation” The different figures are of different quality (fig. 5 shows excellent resolution, while figs. 3 / 4 show limited resolution…). In l. 134 you say that curine significantly reduced the expression of iNOS. Can you comment on how you established significance from the Western blot data?Author Response
Please see the attachment

Round 2
Reviewer 1 Report
The manuscript has been revised accordingly. However, why did the author edit Figures 1 and 2 in the revised manuscript. The error bars are very big. Please check them.
Author Response
Point 1: The manuscript has been revised accordingly. However, why did the author edit Figures 1 and 2 in the revised manuscript? The error bars are very big. Please check them.
Response 1: Figures 1 and 2 have been changed in response to reviewer 2 requests in the first round. In the first version of the manuscript, the results were presented as Means ± SD. However, Reviewer 2 requested that in figures 1 and 2 the results were expressed as Means ± SEM (because we used different numbers of mice per group), which in fact increased the size of the error bars. The number of animals per group above each bar was also entered in response to reviewer 2 requests. However, we believe that this change does not affect the interpretation of the results, since statistical significance, as well as the number of animals in each, was defined in order to improve the presentation of the data, as suggested by reviewer 2.
Reviewer 2 Report
Having addressed all the issues raised by the reviewers, the authors significantly improved their manuscript, which is now suitable for publication.
Author Response
Point 1: Having addressed all the issues raised by the reviewers, the authors significantly improved their manuscript, which is now suitable for publication.
Response 1: Thank you so much for your contribution.